# Acceptance of COVID-19 vaccination in a sample of Middle Eastern and North African (MENA) Houston residents

**Fatin Atrooz**[1], **Zahra Majd**[2], **Ghalya Alrousan**[1], **Sarah Zeidat**[1], **Nadia N. Abuelezam**[3], **Susan Abughosh**[2], **Samina Salim**[1]*

**1** Department of Pharmacological and Pharmaceutical Sciences, College of Pharmacy, University of Houston, Houston, Texas, United States of America, **2** Pharmaceutical Health Outcomes and Policy, College of Pharmacy, University of Houston, Houston, Texas, United States of America, **3** William F. Connell School of Nursing, Boston College, Boston, Massachusetts, United States of America

* ssalim2@central.uh.edu

## Abstract

**Data Availability Statement:** The data is available within the paper.

### Background

COVID-19 vaccination has averted a significant number of deaths in the United States, but vaccination hesitancy continues to be a problem. Therefore, examining vaccination acceptance and/or hesitancy in local communities is critical.

### Methods

A quantitative survey and a multivariable logistic regression model was utilized to determine predictors of COVID-19 vaccination in Middle Eastern and North African (MENA) origin Houston residents. The outcome of interest was COVID-19 vaccination status (vaccinated versus not vaccinated). Covariates included: demographics, health, and COVID-19 factors. Statistical analyses included SAS version 9.4 at a priori significance level of 0.05.

### Results

The overall vaccine acceptance rate was significantly high in this population subset (N = 366), with 77.60% vaccinated, and 22.40% not vaccinated. MENA individuals with some college degrees were less likely to report vaccination than those with a graduate degree [Odds Ratio (OR): 0.18; 95% Confidence Interval (CI): 0.04, 0.77]. Homeowners were more likely to get vaccinated than renters (OR: 2.58; 95%CI: 1.17, 5.68). Individuals practicing Islamic faith were more likely to get vaccinated than other religious affiliations (OR: 3.26; 95%CI: 1.15, 9.19). Individuals with hypertension were less likely to get vaccinated than those without it (OR: 0.34; 95%CI: 0.13, 0.92), and those with anxiety were more likely to get vaccinated than those without anxiety (OR: 4.23; 95%CI: 1.68, 10.64).

**Funding:** This work was supported by Houston Global Health Collaborative Student Research and Service Grant (HGHC211000) awarded to SZ and Grants to Enhance Research on Racism (Grant#000180963), awarded to SS. The funders had no role in study design, data collection and analysis, decision to publish or preparation of the manuscript.

**Competing interests:** The authors have declared that no competing interests exist.

## Conclusions

Health status, education level, financial stability, and religious affiliation are some of the determining factors that potentially influence vaccination acceptance/hesitancy among the MENA community.

## Introduction

The coronavirus disease 2019 (COVID-19), was first detected in the Wuhan City, China in late December 2019 [1]. By the end of January 2020, the disease spread across China and then to many countries including the United States. On March 11, 2020, the World Health Organization (WHO) declared the COVID-19 a global pandemic [2]. The rapid spread of the disease has caused substantial burden on morbidity and mortality, with more than 641 million confirmed cases, and amounting to around 6.6 million deaths worldwide [3]. Vaccination, combined with the adoption of protective measures like wearing masks and social distancing, are considered the most effective strategies to limit the spread of the disease [4]. Given the critical role of vaccination in controlling the pandemic, it is important to examine factors associated with acceptance and/or hesitancy towards vaccination, among different populations [5]. Some studies conducted in high-income countries show that ethnic minorities including migrant populations, may be more reluctant than others in accepting COVID-19 vaccines [6, 7]. A wide variety of ethnicities, cultures and languages represent a broad range of socioeconomic status, educational level, occupation, and English proficiency levels among immigrants settled in the United States (US) [8, 9]. In general, each group has unique cultural, religious, and social practices, and also face healthcare-related barriers, which during the COVID-19 pandemic, may have exacerbated the situation, resulting in disproportionate impact from the COVID-19 pandemic [8]. Many immigrant communities especially recent immigrants and refugees to the US, reportedly experience barriers to vaccination, resulting in lower vaccination rates than US born individuals [10, 11]. This may worsen during public health emergencies such as the current pandemic; thus, hurdles to vaccination among immigrant groups are significant.

Although migration from the Middle East North Africa (MENA) region to the US initiated in the late 1800s, it has accelerated in recent years, driven largely by political turmoil in the region and search for better living conditions and economic opportunities. According to the data provided by the Migration Policy Institute (MPI), in 2016 [12], nearly 3.1 million immigrants from the MENA region lived in the US, accounting for roughly 3 percent of the country's approximately 44 million immigrants [13]. The MENA population in the US is diverse, with approximately 70 percent of MENA immigrants from the Middle East, and around 30 percent from North Africa [13].

Texas has the fourth largest MENA population in the country with over 281,000 MENA individuals living in Texas and approximately 98,300 residing in the Houston metro area [14]. Underreporting of MENA members has been suggested by many policy research groups since the Census Bureau defines "White' as those of European, Middle Eastern or North African descent, thus limiting accurate representation of MENA residents [15, 16]. According to the Arab American Institute, the largest population of MENA Texans resides in the Harris County area, with significant number of MENA Texans living in the Fort Bend County including Katy and Sugar Land [15]. The present COVID-19 survey was therefore targeted towards Harris and Fort Bend County MENA residents.

In the present study, we have examined COVID-19 vaccine hesitancy and acceptance in a sample of MENA group settled in the greater Houston area. This aspect has not been systematically studied in this group, in which religious considerations, cultural practices, misinformation and lack of awareness, may serve as barriers for vaccine acceptance.

## Materials and methods

This was a cross-sectional study based on surveys conducted between July-August 2021 among MENA members residing in the Houston area. We followed the methods as used in our previously published study [17].

### Institutional review board statement

All communication forms and survey questionnaires utilized in the study were approved by the Institutional Review Board (IRB) Committee (STUDY00003078) for the Protection of Human Subjects, University of Houston (UH), Houston, TX, US. The study was conducted according to the guidelines of the Declaration of United States and approved by the Institutional Review Board (or Ethics Committee) of University of Houston (STUDY00003078, 14 May 2020). *Informed Consent Statement*: Informed consent was obtained from all subjects involved in the study.

### Sample size calculation

The minimum sample size was calculated using the G-power 3.1 statistical software [18]. As mentioned earlier, immigrants from MENA countries are severely undercounted because they do not have a well-defined category to denote their background on census surveys. Based on a report published in Houston Chronicle [14], there are around 98,300 MENA residents in Houston area. Therefore, the minimum sample size required to provide 80% power to detect 0.15 effect size for a chi-square analysis and 1.5 odds ratio in a logistic regression analysis at a 0.05 $\alpha$-level was 242 and 308, respectively.

### Survey

The survey comprised of previously validated questions that were originally written in English [19] then translated into Arabic by a bilingual Arabic translator [20]. The quantitative survey questions consisted of sociodemographic, general health, and COVID-19 specific questions. The sociodemographic questions comprised of 10 questions that were related to age, gender, race/ethnicity, country of origin, religion, citizenship status, marital status, educational level, living status, and annual household income. The COVID-19 related questions included: prior COVID-19 infection, vaccination status, practices of protective measures, perceived infection risk and severity, and perceived knowledge about self-protection. Health questions included: health insurance status, overall health status, smoking status, alcohol consumption status, and disease comorbidities including hypertension, hypercholesterolemia, obesity, and anxiety. The survey questionnaire was uploaded on REDCap platform in both languages, English and Arabic. Arabic is the most spoken native language among the MENA community. Other languages spoken by the MENA community are Persian or Farsi, Hebrew, and Turkish. The link for the survey was sent to MENA community through Houston-based 501©(3) non-profit organization Multi Cultural Center (MCC), Webster, TX (http://www.multiculturalcenter.net/). *Inclusion Criteria*: MENA individuals 18 years old and above, who are settled in the Houston area were included in this study. MENA group included those who identified their country of origin as one of MENA countries. *Exclusion criteria*: Non-members of the MENA group, MENA

members residing out of Houston area, and MENA members under the age of 18 were excluded.

## Statistical analysis

All statistical analyses were performed using SAS version 9.4 (SAS Institute, Cary, NC) at a priori significance level of 0.05. The participants' sociodemographic, self-reported general health, and responses to COVID-19 related questions were summarized using descriptive analysis. The frequencies for all categorical variables were calculated and described as percentages. Group differences were assessed using chi-square tests. A multivariable logistic regression model was conducted to determine predictors of receiving COVID-19 vaccination. The outcome of interest was COVID-19 vaccination status: "vaccinated" vs "not vaccinated". Covariates included in the model were gender, age, education, marital status, living status, annual household income, religion, smoking status, alcohol consumption status, health insurance status, general health self-evaluation, and presence of chronic diseases such as hypertension, hypercholesterolemia, obesity, and anxiety. COVID-19 related factors included prior infection, perceived infection risk and severity, perceived knowledge on self-protection from the COVID-19 infection and ability to avoid infection. Variables such as presence of cancer, heart attack, stroke, asthma, dementia, arthritis, and chronic obstructive pulmonary disease were excluded as a very small number of the survey respondents reported having any of these conditions.

## Results

### Demographic characteristics

Table 1 represents the sociodemographic characteristics of the 368 survey respondents. A total of 409 individuals clicked on the survey link and initiated the survey with 368 (89.9%) completing more than 90% of the survey questions. A total of 366 individuals were included in this analysis with few-missing data on vaccine intention, age, gender, education, health insurance, income, employment, citizenship, and religiosity. Data collection started on July 21, 2021 and ended on August 01, 2021. The respondents included 243 males (66.39%). Majority of the respondents (67.76%) were within the age group 26–39 years. Most of the participants reported to have attended some college (27.60%) and 48.09% reporting attainment of college degrees. The average annual income for most of the respondents (46.72%) was between $45,000- $100,000. (Table 1). *Overall health*: Most respondents (65.30%) stated very good/ good self-reported health. Some of the respondents reported the following health issues: hypertension (14.75%), high cholesterol (13.11%), and obesity (14.21%). Around one third (28.42%) of the respondents reported having anxiety issues (Table 2). *COVID-19 infection*: In this sample, 69 individuals (18.85%) reported having had a COVID-19 infection, which is very close to the percentage of those reporting severe COVID-19 infection (17.49%). (Table 2).

### Multivariable logistic regression analysis

We observed that MENA individuals with college degree had lower odds of reporting vaccination when compared to individuals with a graduate degree [Odds Ratio (OR): 0.18; 95% Confidence Interval (CI): 0.04, 0.77]. Homeowners had greater odds of reporting vaccination when compared to those residing in rented properties (OR: 2.58; 95% CI: 1.17, 5.68). Those practicing the Islamic faith (Muslims) had higher odds of reporting vaccination compared to individuals practicing other religions. (OR: 3.26; 95% CI: 1.15, 9.19). Those with hypertension had lower odds of reporting vaccination when compared to those without hypertension (OR: 0.34;

**Table 1. Baseline demographic characteristics and COVID-19 vaccination.**

| | Total (%) | Vaccinated for COVID19 | | p Value |
|---|---|---|---|---|
| | | No | Yes | |
| **Variable** | | | | |
| Gender | | | | |
| Males | 243 (66.39) | 59 (71.95) | 184 (64.79) | 0.2265 |
| Females | 123 (33.61) | 23 (28.05) | 100 (35.21) | |
| Age (years) | | | | |
| 18–25 | 85 (23.22) | 14 (17.07) | 71(25.00) | 0.2961 |
| 26–39 | 248 (67.76) | 61 (74.39) | 187(65.85) | |
| ≥40 | 33 (9.02) | 7 (8.54) | 26 (9.15) | |
| Education | | | | |
| High school or less | 50 (13.66) | 9 (10.98) | 41 (14.44) | 0.0022* |
| Some college or associate degree | 101(27.60) | 36 (43.90) | 65 (22.89) | |
| College degree | 176 (48.09) | 32 (39.02) | 144 (50.70) | |
| Graduate degree | 39 (10.66) | 5 (6.10) | 34 (11.97) | |
| Marital Status | | | | |
| Married | 246 (67.21) | 53 (64.63) | 193 (67.96) | 0.7509 |
| Single | 79 (21.58) | 18 (21.95) | 61 (21.48) | |
| Divorced/Separated | 41(11.20) | 11 (13.41) | 30 (10.56) | |
| Living Status | | | | |
| Own | 251(68.58) | 48 (58.54) | 203 (71.48) | 0.0261* |
| Rent | 115 (31.42) | 34 (41.46) | 81 (28.52) | |
| Annual Household Income | | | | |
| <$25,000 | 59 (16.12) | 10 (12.20) | 49 (17.25) | 0.7224 |
| $25,000 to < $45,000 | 94 (5.68) | 21 (25.61) | 73 (25.70) | |
| $45,000 to < $65,000 | 97 (26.50) | 25 (30.49) | 72 (25.35) | |
| $65,000 to <$100,000 | 74 (20.22) | 18 (21.95) | 56 (19.72) | |
| ≥ $100,000 | 42 (11.48) | 8 (9.76) | 34 (11.97) | |
| Religion | | | | |
| Christian | 126 (34.43) | 27 (32.93) | 99 (34.86) | 0.0947 |
| Muslim | 194 (53.01) | 39 (47.56) | 155 (54.58) | |
| Jewish/other | 46 (12.57) | 16 (19.51) | 30 (10.56) | |
| Smoke | | | | |
| No | 222 (60.66) | 58 (70.73) | 164 (57.75) | 0.0340* |
| Yes | 144 (39.34) | 24 (29.27) | 120 (42.25) | |
| Drink Alcohol | | | | |
| No | 251 (68.58) | 59 (71.95) | 192 (67.61) | 0.4552 |
| Yes | 115 (31.42) | 23 (28.05) | 92 (32.39) | |
| Health Insurance | | | | |
| Yes | 291(79.51) | 55 (67.07) | 236 (83.10) | 0.0015* |
| No | 75 (20.49) | 27 (32.93) | 48 (16.90) | |

*Significant p value from chi square <0.05

95%CI: 0.13, 0.92). Those with anxiety had higher odds of reporting vaccination when compared to those who reported no anxiety. (OR: 4.23; 95%CI: 1.68, 10.64). Other covariates included in the model (such as gender and age) were not correlated with vaccination status (Table 3).

**Table 2. Baseline health characteristics and COVID-19 vaccination.**

| | Total (%) | Vaccinated for COVID19 | | p Value |
|---|---|---|---|---|
| | | No | Yes | |
| **Variable** | | | | |
| Overall Health | | | | |
| Excellent | 105 (28.69) | 16 (19.51) | 89 (3134) | 0.0487* |
| Very Good/Good | 239 (65.30) | 58 (70.73) | 181(63.73) | |
| Fair/Poor | 22 (6.01) | 8 (9.76) | 14 (4.93) | |
| Hypertension | | | | |
| Yes | 54 (14.75) | 16 (19.51) | 38 (13.38) | 0.3848 |
| No | 302 (82.51) | 64 (78.05) | 238 (83.80) | |
| Unknown | 10 (2.73) | 2 (2.44) | 8 (2.82) | |
| High Cholesterol | | | | |
| Yes | 48 (13.11) | 9 (10.98) | 39 (13.73) | 0.2304 |
| No | 306 (83.61) | 68 (82.93) | 238 (83.80) | |
| Unknown | 12 (3.28) | 5 (6.10) | 7 (2.46) | |
| Obesity | | | | |
| Yes | 52 (14.21) | 12 (14.63) | 40 (14.08) | 0.8411 |
| No | 296 (80.87) | 65 (79.27) | 231 (81.34) | |
| Unknown | 18 (4.92) | 5 (6.10) | 13 (4.58) | |
| Anxiety | | | | |
| Yes | 104 (28.42) | 11 (13.41) | 93 (32.75) | 0.0029** |
| No | 248 (67.76) | 67 (81.71) | 181 (63.73) | |
| Unknown | 14 (3.83) | 4 (4.88) | 10 (3.52) | |
| Infected with COVID-19 | | | | |
| Yes | 69 (18.85) | 12 (14.63) | 57 (20.07) | 0.3921 |
| No | 295 (80.60) | 70 (85.37) | 225 (79.23) | |
| Unknown | 2 (0.55) | 0 (0.00) | 2 (0.70) | |
| COVID-19 Infection Risk | | | | |
| Extremely Likely | 41 (11.20) | 5 (6.10) | 36 (12.72) | 0.3754 |
| Somewhat Likely | 145 (39.62) | 34 (41.46) | 111 (39.22) | |
| Neither Likely nor Unlikely/Somewhat or Extremely Unlikely | 179 (49.91) | 43 (52.44) | 136 (48.06) | |
| Unknown | 1 (0.27) | 0 (0.00) | 1 (0.35) | |
| COVID-19 Infection Severity | | | | |
| Extremely Severe | 64 (17.49) | 12 (14.63) | 52 (18.31) | 0.4305 |
| Somewhat Severe | 156 (42.62) | 41 (50.00) | 115 (40.49) | |
| Neither Severe nor Mild | 95 (25.96) | 20 (24.39) | 75 (26.41) | |
| Somewhat/Extremely Mild | 45 (12.30) | 9 (10.98) | 36 (12.68) | |
| Unknown | 6 (1.64) | 0 (0.00) | 6 (2.11) | |
| Know Self-Protection from COVID-19 Infection | | | | |
| Strongly Agree | 123 (33.61) | 19 (23.17) | 104 (36.62) | 0.0487* |
| Somewhat Agree | 169 (46.17) | 41 (50.00) | 128 (45.07) | |
| Neither Agree nor Disagree/Somewhat Disagree | 74 (20.22) | 22 (26.83) | 52 (18.31) | |
| Avoiding COVID-19 Infection | | | | |
| Extremely Easy/Somewhat Easy | 200 (54.64) | 48 (58.54) | 152 (53.52) | 0.8601 |
| Neither Easy nor Difficult | 111 (30.33) | 22 (26.83) | 89 (31.34) | |
| Somewhat/Extremely Difficult | 51(13.93) | 11 (13.41) | 40 (14.08) | |
| Unknown | 4 (1.09) | 1 (1.22) | 3 (1.06) | |

*Significant p value from chi square <0.05

**Significant p value from chi square <0.001

**Table 3. Multivariable logistic regression of COVID-19 vaccination.**

| Variable | Vaccinated vs Not Vaccinated | | | |
|---|---|---|---|---|
| | Unadjusted OR (95% CI) | p Value | Adjusted OR (95% CI) | p Value |
| Gender | | | | |
| Male vs Female | 0.71 (0.41, 1.23) | 0.2277 | 0.84 (0.41, 1.71) | 0.6322 |
| Age | | | | |
| 18–25 vs ≥40 | 1.36 (0.49, 3.75) | 0.2698 | 2.26 (0.52, 9.82) | 0.1944 |
| 26–39 vs ≥40 | 0.82 (0.34, 1.99) | 0.2424 | 1.23 (0.36, 4.17) | 0.6292 |
| Education | | | | |
| Less than High School vs Graduate Degree | 0.67 (0.20, 2.18) | 0.6641 | 0.61 (0.11, 3.38) | 0.5966 |
| Some College/Associate Degree vs Graduate Degree | 0.26 (0.09, 0.73) | 0.0004* | 0.18 (0.04, 0.77) | 0.0019* |
| College Degree vs Graduate Degree | 0.66 (0.24, 1.82) | 0.5735 | 0.46 (0.11, 1.87) | 0.9179 |
| Marital Status | | | | |
| Married vs Widowed /Separated | 1.33 (0.62, 2.84) | 0.5043 | 0.94 (0.31, 2.90) | 0.7651 |
| Single vs Widowed/Separated | 1.24 (0.52, 2.96) | 0.8258 | 0.7 (0.20, 2.38) | 0.4748 |
| Living Status | | | | |
| Own vs Rent | 1.77 (1.06, 2.95) | 0.0272 | 2.58 (1.17, 5.68) | 0.0190* |
| Annual Household Income | | | | |
| <$25,000 vs ≥ $100,000 | 1.15 (0.41, 3.22) | 0.3291 | 2.96 (0.68, 12.81) | 0.3457 |
| $25,000 to < $45,000 vs ≥ $100,000 | 0.81 (0.32, 2.03) | 0.8351 | 3.16 (0.81, 12.25) | 0.1921 |
| $45,000 to < $65,000 vs ≥ $100,000 | 0.67 (0.27, 1.65) | 0.2931 | 1.74 (0.51, 5.93) | 0.6488 |
| $65,000 to <$100,000 vs ≥ $100,000 | 0.73 (0.28, 1.86) | 0.5227 | 1.97 (0.53, 7.36) | 0.9643 |
| Religion | | | | |
| Christian vs Jewish/Other | 1.95 (0.93, 4.10) | 0.2942 | 2.28 (0.79, 6.57) | 0.5456 |
| Muslim vs Jewish/Other | 2.12 (1.05, 4.27) | 0.1103 | 3.26 (1.15, 9.19) | 0.0429* |
| Smoke | | | | |
| No vs Yes | 0.56 (0.33, 0.96) | 0.0353* | 0.83 (0.39, 1.76) | 0.6224 |
| Drink Alcohol | | | | |
| No vs Yes | 0.81 (0.47, 1.39) | 0.4557 | 0.64 (0.29, 1.41) | 0.2721 |
| Health Insurance | | | | |
| Yes vs No | 2.41 (1.38, 4.20) | 0.0019* | 2.06 (0.94, 4.48) | 0.0696 |
| Overall Health | | | | |
| Excellent vs Fair/Poor | 3.17 (1.14, 8.80) | 0.0156* | 3.68 (0.86, 15.64) | 0.0533 |
| Very Good/Good vs Fair/Poor | 1.78 (0.71, 4.46) | 0.9994 | 2.02 (0.54, 7.60) | 0.8998 |
| Hypertension | | | | |
| Yes vs No | 0.63 (0.33, 1.21) | 0.1734 | 0.34 (0.13, 0.92) | 0.0340* |
| High Cholesterol | | | | |
| Yes vs No | 1.23 (0.57, 2.68) | 0.5883 | 1.9 (0.57, 6.27) | 0.2952 |
| Obesity | | | | |
| Yes vs No | 0.93 (0.46, 1.89) | 0.8579 | 1.3 (0.47, 3.62) | 0.6152 |
| Anxiety | | | | |
| Yes vs No | 3.12 (1.57, 6.20) | 0.0011* | 4.23 (1.68, 10.64) | 0.0022* |
| Infected with COVID-19 | | | | |
| Yes vs No | 1.47 (0.75, 2.91) | 0.2591 | 1.03 (0.42, 2.53) | 0.9555 |
| COVID-19 Infection Risk | | | | |
| Extremely Likely vs Neither Likely nor Unlikely/ Somewhat Unlikely | 2.27 (0.84, 6.16) | 0.1031 | 0.94 (0.24, 3.74) | 0.9125 |
| Somewhat Likely vs Neither Likely nor Unlikely/ Somewhat Unlikely | 1.03 (0.61, 1.72) | 0.2369 | 1.04 (0.51, 2.10) | 0.8838 |
| COVID-19 Infection Severity | | | | |
| Extremely Severe vs Somewhat Mild/Extremely Mild | 1.08 (0.41, 2.83) | 0.5401 | 0.81 (0.22, 3.02) | 0.8013 |

*(Continued)*

**Table 3.** (Continued)

| Variable | Unadjusted OR (95% CI) | p Value | Adjusted OR (95% CI) | p Value |
|---|---|---|---|---|
| | **Vaccinated vs Not Vaccinated** | | | |
| Somewhat Severe vs Somewhat Mild/Extremely Mild | 0.70 (0.31, 1.58) | 0.164 | 0.66 (0.22, 2.02) | 0.2668 |
| Neither Severe nor Mild vs Somewhat Mild/Extremely Mild | 0.93 (0.38, 2.26) | 0.9294 | 1.19 (0.37, 3.86) | 0.3878 |
| Know Self-Protection from COVID-19 Infection | | | | |
| Strongly Agree vs Neither Agree nor Disagree/Somewhat Disagree/Strongly Disagree | 2.31 (1.15, 4.65) | 0.0172* | 2.68 (0.90, 8.04) | 0.0932 |
| Somewhat Agree vs Neither Agree nor Disagree/Somewhat Disagree/Strongly Disagree | 1.32 (0.71, 2.43) | 0.5754 | 1.63 (0.66, 4.02) | 0.9835 |
| Avoiding COVID-19 Infection | | | | |
| Extremely/Somewhat Easy vs Somewhat/Extremely Difficult | 0.87 (0.41, 1.82) | 0.4707 | 0.98 (0.36, 2.64) | 0.2586 |
| Neither Easy nor Difficult vs Somewhat/Extremely Difficult | 1.11 (0.49, 2.51) | 0.5634 | 2.2 (0.737, 6.56) | 0.0566 |

*Significant $p$ value from chi square $<0.05$

## Discussion

Research on health behaviors especially vaccination trends among MENA population in the US lags as compared to other immigrant and minority groups [21–23]. Only a few studies have examined COVID-19 vaccine hesitancy among Arab Americans [20, 24, 25, 26], with the focus areas primarily being Michigan, Minnesota, California, New York and Virginia as the most frequent research sites for examining MENA health trends [27]. This cross-sectional study aimed at understanding the predictors of COVID-19 vaccination among MENA individuals focused on Harris County and Fort Bend MENA residents. There is paucity of research in this demographic in Texas and to our knowledge this is the first study investigating MENA COVID-19 vaccination acceptance in Houston, Texas, which is a follow up of our previous publication in which we had investigated the perception regarding knowledge of COVID-19 prevention among the MENA community [17].

In this pilot study, the overall vaccine acceptance within a sample of Houston MENA population was high, with 77.60% of respondents reported being vaccinated for COVID-19. According to the Harris County commissioner's office 48.2% of Harris County residents [28] and 43.9% Fort Bend County (Fort Bend County-coronavirus [29] residents were fully vaccinated as of August 1, 2021. Thus, vaccine acceptance was high among our sample of Harris and Fort Bend County MENA residents, when compared to overall vaccination status of residents in these two counties. Our findings are consistent with previous study conducted in a national sample of Arab Americans wherein 92% of respondents reported having received at least one dose of a COVID-19 vaccine [24]. The high vaccination rates in MENA individuals reported in our study and also in other studies conducted in the US highlight the stark difference in the vaccination rate reported in the MENA countries, where the highest rate was reported in Tunisia (92%), and the lowest rate was reported in Iraq (13%) [30, 31]. Vaccine hesitancy in MENA population is reportedly influenced by many factors, such as health status, educational level, financial stability and religious affiliations [32–34]. Misinformation such as inclusion of pork products in vaccines or purported risk of autism and cancer, healthcare access and system familiarity, limited trust of government, and specific vaccine concerns of efficacy, necessity, and safety, combined, make a vaccine less favorable for acceptance in new immigrants and ethnic minority groups including MENA immigrants [35]. Our findings show that majority of MENA individuals in our sample exhibit very good/good health, however, some reported chronic diseases including hypertension, high cholesterol, obesity, and anxiety.

Of these health issues, anxiety and hypertension were predictors of COVID-19 vaccination status. In general fear and anxiety of contracting the virus, concerns regarding vaccine safety, and financial hardships are well-described [36–38]. MENA individuals in our sample with anxiety were more likely to get vaccinated. This seems plausible considering reported anxiety and fear of acquiring COVID-19 infection referred as "functional fear" is suggested to promote vaccination uptake in many populations [37, 39], while concerns over vaccine safety in general and social media-misinformation concerning vaccination safety has significantly reduced vaccine uptake among individuals with high anxiety [39].

Our findings suggest that MENA individuals with hypertension were less likely to get vaccinated. The comorbidity between hypertension and COVID-19 and the increased mortality were reported in many countries [40, 41], and several studies have reported people with chronic diseases exhibiting lower vaccine acceptance than those without chronic conditions [42]. Some studies have reported that some patients developed hypertension following COVID-19 vaccination [43, 44] adding to the lower vaccine uptake by hypertensive individuals. Educational level also impacted vaccine acceptance in this sample. Respondents with a graduate degree were more likely to get vaccinated when compared to those with college or associate degree. Educational differences were also associated with vaccine acceptance/hesitancy in some countries. Individuals with high levels of education were more likely to accept vaccination in Ecuador, France, Germany, India, and the US. however, higher education was linked to lower vaccine acceptance in Canada, Spain, and the UK [45]. The increased vaccine acceptance among individuals with higher education in this sample might be explained by high awareness and discrediting of conspiracy theories and misinformation regarding COVID-19 vaccination that has been widespread in the social media during the pandemic [46]. Furthermore, MENA immigrants are believed to have high levels of educational attainment in the US, with roughly half of college educated MENA immigrants reportedly hold degrees in STEM fields [47], disciplines that enable a better understanding of vaccine science, thus facilitating better vaccine acceptance.

Our study also reveals the potential impact of economic stability status on vaccine acceptance. Homeowners were more likely to get vaccination compared to those that lived in rented accommodation. Financial instability is associated with decreased vaccine acceptance in many countries [48]. MENA community is reportedly highly affluent in Texas, as the spending power of MENA immigrants reportedly topped $2 billion in 2015, with Houston, Sugarland and Woodland areas combined exhibiting $1.2 billion in spending power of MENA households [47]. Moreover, our regression model revealed that Muslims among the MENA community were more likely to get the vaccines as compared to respondents from other religions. A point of consideration to be made is that more than half of respondents in this sample identified themselves as Muslims (53%). According to a recent poll by the Institute for Social Policy and Understanding (ISPU), Muslims who identified themselves as either Asian or white were more likely to get vaccinated than Muslims who identified as Arab and/or Black (42% of both Asian and white Muslims versus 28% of Arab and 29% of Black Muslims). And Muslims who identified as Black were least likely to have received at least one vaccine dose (3% versus more than 16% for Asian, Arab, and white Muslims) and most likely to say they do not intend to get vaccinated (38% versus less than 15% for Asian, Arab, and white Muslims) [49]. This is not surprising considering medical mistrust among Black Americans due to the racism-related issues [50].

Vaccination hesitancy is declared as a top threat to global health [51]. In the US, 78.5% of all 18 and older are fully vaccinated [52]. Unvaccinated remain the most vulnerable to COVID-19 infection. Findings from this study will help address the factors that contribute to vaccination hesitancy among minorities in the US.

## Conclusion

Our data suggest that hesitancy to COVID-19 vaccination at least within the MENA community is associated with health status, educational level, as well as financial status and identified subpopulations within the MENA community that are likely to benefit from vaccine promotion interventions. This information is significant for informing public health experts to improve effective messaging by incorporating cultural competence and language-tailored dissemination strategies to promote vaccine uptake in targeted subpopulations. Family practitioners and religious leaders can play a greater role by promoting culturally competent and language tailored vaccine safety information among their MENA patients to promote greater vaccine acceptance and prevent COVID-19 infections and subsequent health issues within MENA community.

## Limitations

This study has several limitations. First, given that this study was conducted in a limited urban area (Harris County and Fort Bend County), the results should not be generalized to all of Houston, Texas. Also, while some of the findings may be applicable to the MENA population residing in the US, our findings are not reflective of the entire MENA group or the American population in general, given that our study only examined a small sample of MENA community and included Arabic speaking respondents. Also, our study captured vaccine attitudes at one timeframe and towards COVID-19 virus, therefore, our results cannot be related to an overall vaccine hesitancy within this group. Second, this is a cross sectional study, the possibility that respondents' intention toward vaccination was based on previous history of infection cannot be excluded. Third, the sample size was limited, the fact that the MENA group race/ethnicity is not distinguished and considered as part of white population limited the outreach of targeted people. Fourth, MENA community is a heterogenous group including individuals of Arab, Turkish, and Persian descent, with varying socio-economic status. However, our sample included predominantly Arab members, therefore, COVID-19 vaccination related behaviors among Turkish and Persian community were not discussed. Additionally, our data was based on self-report, hence the responses might have been affected by the respondents' overall state of well-being; exaggeration and various biases may have affected the results. Furthermore, our survey dissemination may not have reached the entire Houston MENA community including MENA refugees, therefore, health disparities often prevalent in refugee community may not have been revealed. Also, direct comparisons between MENA and non-MENA Houston residents are challenging due to lack of a separate MENA category which prevents delineating and comparing accurate health trends of this group with other minorities or ethnic groups. As pointed out in a recent study, the unequal impact of COVID-19 on racial and ethnic minorities in comparison to non-Hispanic whites neglects to account for the potential differences in COVID-19 infection within the non-Hispanic white population. Non-Hispanic whites are heterogeneous and defined by the federal government as persons from Europe, the Middle East, or North Africa [53]. This is a hypothesis generating cross sectional study which was conducted to identify which variables affect the outcome (vaccination). Future studies will build on the present results and test specific independent variables as described. Thus, this survey-based study was hypothesis generating and exploratory in which we explored potential associations with receiving COVID19 vaccination and hence no attempt was made to understand causal inference by including a primary exposure. Given the limitations of convenience sampling, small sample size, and a cross-sectional design, this study did not look at causality. Therefore, future studies are recommended to examine causal associations and provide appropriate adjusted estimates for primary and secondary exposures.

## Future studies

Several attractive possibilities for future studies include; a) examining COVID-19 vaccination acceptance/hesitancy across all Houston counties (Houston has nine counties: Austin, Brazoria, Chambers, Fort Bend, Galveston, Harris, Liberty, Montgomery and Waller); b) examining COVID-19 vaccination acceptance/hesitancy among Houston MENA refugees across all 9 counties; c) designing and adopting culturally competent and language tailored community-level outreach programs to promote awareness and vaccination among MENA community would greatly minimize health inequities and promote public health education among MENA groups at the local, as well as national level. This integrated behavior change strategy will increase acceptance of COVID-19 vaccines.

## Acknowledgments

Authors acknowledge help and support of Dr. Uzma Khan (PharmD), Multi-Cultural Center study coordinator for her help with MENA survey dissemination.

## Author Contributions

**Conceptualization:** Sarah Zeidat, Nadia N. Abuelezam, Susan Abughosh, Samina Salim.

**Data curation:** Fatin Atrooz, Zahra Majd, Sarah Zeidat, Samina Salim.

**Formal analysis:** Fatin Atrooz, Zahra Majd, Ghalya Alrousan.

**Funding acquisition:** Samina Salim.

**Investigation:** Samina Salim.

**Methodology:** Fatin Atrooz, Zahra Majd, Ghalya Alrousan, Nadia N. Abuelezam, Samina Salim.

**Project administration:** Samina Salim.

**Resources:** Samina Salim.

**Software:** Fatin Atrooz.

**Supervision:** Samina Salim.

**Validation:** Fatin Atrooz, Zahra Majd, Samina Salim.

**Writing – original draft:** Fatin Atrooz, Zahra Majd, Sarah Zeidat.

**Writing – review & editing:** Nadia N. Abuelezam, Susan Abughosh, Samina Salim.

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
