## [Decision Letter · Decision Letter 0]

1 Dec 2022

PGPH-D-22-01593

Acceptance of COVID-19 Vaccination in a Sample of Middle Eastern and North African (MENA) Houston Residents

Dear Dr. Salim,

Thank you for submitting your manuscript to PLOS Global Public Health. After careful consideration, we feel that it has merit but does not fully meet PLOS Global Public Health’s publication criteria as it currently stands. Therefore, we invite you to submit a revised version of the manuscript that addresses the points raised during the review process.

We look forward to receiving your revised manuscript.

Kind regards,

Kathleen Bachynski, PhD, MPH

Academic Editor

Journal Requirements:

2. Please send a completed 'Competing Interests' statement, including any COIs declared by your co-authors. If you have no competing interests to declare, please state "The authors have declared that no competing interests exist". Otherwise please declare all competing interests beginning with the statement "I have read the journal's policy and the authors of this manuscript have the following competing interests:"

3. Please amend your detailed Financial Disclosure statement. This is published with the article. It must therefore be completed in full sentences and contain the exact wording you wish to be published.

a. State what role the funders took in the study. If the funders had no role in your study, please state: “The funders had no role in study design, data collection and analysis, decision to publish, or preparation of the manuscript.”

b. If any authors received a salary from any of your funders, please state which authors and which funders.

4. Please ensure that Funding Information and Financial Disclosure Statement are matched

5. In the online submission form, you indicated that "Data can be made available upon request". All PLOS journals now require all data underlying the findings described in their manuscript to be freely available to other researchers, either 1. In a public repository, 2. Within the manuscript itself, or 3. Uploaded as supplementary information. This policy applies to all data except where public deposition would breach compliance with the protocol approved by your research ethics board. If your data cannot be made publicly available for ethical or legal reasons (e.g., public availability would compromise patient privacy), please explain your reasons by return email and your exemption request will be escalated to the editor for approval. Your exemption request will be handled independently and will not hold up the peer review process, but will need to be resolved should your manuscript be accepted for publication. One of the Editorial team will then be in touch if there are any issues.

Additional Editor Comments (if provided):

Thank you very much for submitting your manuscript to PLOS Global Health. The manuscript addresses an important area of global health research. Both reviewers highlighted the important contributions that this study makes to the literature. The reviewers recommended some minor revisions; therefore, I invite you to respond to the reviewers’ comments and revise the manuscript.

Reviewers' comments:

Reviewer's Responses to Questions

**Comments to the Author**

1. Does this manuscript meet PLOS Global Public Health’s publication criteria? Is the manuscript technically sound, and do the data support the conclusions? The manuscript must describe methodologically and ethically rigorous research with conclusions that are appropriately drawn based on the data presented.

Reviewer #1: Yes

Reviewer #2: Yes

2. Has the statistical analysis been performed appropriately and rigorously?

Reviewer #1: Yes

Reviewer #2: Yes

3. Have the authors made all data underlying the findings in their manuscript fully available (please refer to the Data Availability Statement at the start of the manuscript PDF file)?

Reviewer #1: No

Reviewer #2: Yes

4. Is the manuscript presented in an intelligible fashion and written in standard English?

Reviewer #1: Yes

Reviewer #2: Yes

5. Review Comments to the Author

Reviewer #1: The authors have done stellar work in this paper. I offer a few suggestions meant to strengthen the manuscript:

1. The introduction is very long and can be shortened. Consider starting with a brief overview of the research that has been published about Covid-19 vaccination hesitancy and how this research has focused on minorities and whites. Then transition to the fact that MENA are identified as white. The section about misinformation can be moved into the discussion section. You should retain the data about MENA in Texas. The paragraph on page 5 (lines 104-113) can be integrated to the discussion section.

2. On page 6, under the Survey section, you do not need to name the person who developed the survey. I would suggest you write something like "one of the co-authors."

3. On page 7, line 144 needs a reference after you mention the survey that was developed. I am curious: were some of the survey questions borrowed from other surveys about Covid-19 vaccine hesitancy? Either way, this should be stated.

4. When describing the tables, I suggest the authors highlight a few of the findings. There is no need to describe each finding, especially in tables 1 and 2.

5. In the multivariable logistic regression model, I suggest the authors present a crude, unadjusted odds ratio and confidence intervals and then add variables based on grouping, such as demographics, health behaviors, health conditions, Covid-19 characteristics. I also suggest the authors present a main independent variable versus including all of the covariates to see which ones have statistically significant influence. For example, based on previous research and knowledge about the MENA community, there may be a reason to believe that Covid-19 characteristics (infected, risk, severity, self-protection, avoidance) are the main predictors of vaccine hesitancy. So, the authors would present crude odds ratios for those (or even create a score of the Covid-19 characteristics) and then add "groupings" of demographic, health behaviors, etc., that may influence the crude estimates. I think this presentation will make the analysis more methodical and justified.

6. Lastly, I urge you to search for recently published articles about Covid-19 vaccine hesitancy among MENA -- there might have been some publications since the submission of your manuscript.

Reviewer #2: In the current study, authors have investigated the acceptance of COVID-19 vaccination in a Sample of Middle Eastern and North African in Texas, Houston area. The manuscript is well-written and informative. I recommend acceptance of the manuscript. I have minor comments as indicated below:

In the Introduction line 59, Reference 44 is out of order. It should labelled as Ref 3. Same applies to Ref 44 line 102, Ref 48 line 138.

Referencing issues are also noted in lines: 249, 231, 247

In the Introduction line 77, replace United State with US

In the Introduction line 87-88, MENA should be defined in the first use

In the Methods, ethical approval is repeated twice in line 122 and in Line 129-132.

Line 192: change “Obesity” to “obesity”.

In the Discussion, I suggest that authors to compare their results to some Data from MENA countries. For Example: PMID: 36394768, PMID: 34014128 and PMID: 35857813

6. PLOS authors have the option to publish the peer review history of their article (what does this mean?). If published, this will include your full peer review and any attached files.

**Do you want your identity to be public for this peer review?** For information about this choice, including consent withdrawal, please see our Privacy Policy.

Reviewer #1: **Yes: **Florence Jamil Dallo

Reviewer #2: No

---

## [Editor Report · Decision Letter 1]

9 Jan 2023

Acceptance of COVID-19 Vaccination in a Sample of Middle Eastern and North African (MENA) Houston Residents

PGPH-D-22-01593R1

Dear Dr Salim,

We are pleased to inform you that your manuscript 'Acceptance of COVID-19 Vaccination in a Sample of Middle Eastern and North African (MENA) Houston Residents' has been provisionally accepted for publication in PLOS Global Public Health.

Best regards,

Kathleen Bachynski, PhD, MPH

Academic Editor

My thanks to the reviewers for their constructive recommendations and to the authors for their revisions and responses to the reviewer comments.